EMBO
Molecular Medicine

# SMAD7: riding on fibrosis-limiting routes and beyond

Leonardo Martin [1,2✉], Giulio Gabbiani[3] & Guido R Y De Meyer [1,2]

## Abstract

**Fibrosis, marked by excessive extracellular matrix deposition, underlies the progression of major organ pathologies, including cardiac and skeletal muscle diseases. Central to fibrotic remodeling is the persistent activation of myofibroblasts, orchestrated by profibrotic mediators such as transforming growth factor-beta (TGF-β). SMAD7, a key inhibitor of TGF-β signaling, has emerged as both an antifibrotic effector and a modulator of immune and tissue remodeling responses. New insights reveal that SMAD7 exerts cell-specific antifibrotic effects, particularly within myofibroblasts, limiting macrophage-driven fibrogenesis through paracrine mechanisms. Moreover, the integration of SMAD7 modulation into engineered cellular therapies, such as CAR-T cells, highlights its potential to enhance regenerative outcomes and immune resilience against fibrosis. Here, we review the expanding role of SMAD7 in cardiac, skeletal muscle, and vascular tissues, emphasizing its promise as a therapeutic target for reprogramming fibrosis, promoting tissue repair, and restoring organ function in chronic disease settings.**

## Background

Fibrosis, defined by excessive extracellular matrix (ECM) deposition due to dysregulated wound healing and tissue repair, affects multiple organs, including the heart, liver, muscle, lung, and kidney. It plays a central role in conditions such as myocardial infarction and post-injury muscle repair, where it significantly contributes to disease progression. Persistent activation of myofibroblasts—driven by profibrotic mediators like transforming growth factor-beta (TGF-β), platelet-derived growth factor (PDGF), and epidermal growth factor (EGF)—is a hallmark of fibrotic remodeling. In tissues with limited regenerative capacity, such as the adult myocardium, fibrosis initially serves a reparative purpose by preserving structural integrity and preventing complications like ventricular rupture. However, when fibrosis becomes excessive or unresolved, it drives maladaptive remodeling, myocardial stiffening, impaired contractility, and ultimately heart failure. Tightly regulated fibrotic activity is therefore essential for maintaining cardiac homeostasis.

TGF-β signaling is transduced primarily through SMAD transcription factors, which include receptor-activated SMADs (R-SMADs), the common mediator SMAD4, and inhibitory SMADs (I-SMADs). Among the latter, SMAD7 functions as a key negative regulator by disrupting downstream TGF-β signaling at multiple levels.

While SMAD7 has been widely studied in liver and kidney fibrosis—where it dampens TGF-β activity and restrains inflammation-driven ECM accumulation—recent evidence highlights its relevance in cardiac, skeletal muscle, and vascular fibrosis. In these mechanically dynamic tissues, SMAD7 integrates biomechanical and inflammatory cues to fine-tune fibrotic remodeling.

In this context, SMAD7 emerges as a pivotal modulator of fibrosis, particularly in striated muscle and cardiovascular systems. By antagonizing both canonical and noncanonical TGF-β pathways, it offers a promising therapeutic avenue for limiting fibrosis while preserving essential repair mechanisms. This Perspective summarizes recent insights into SMAD7's roles in tissue regeneration and immune modulation, and explores its potential in targeted antifibrotic strategies.

## The transformation of fibroblasts into myofibroblasts: processes and underlying mechanisms

Fibroblasts, which are ubiquitous in human tissues, play a critical role in maintaining the structural integrity and function of these tissues. They are responsible for synthesizing the extracellular matrix (ECM), a three-dimensional network of macromolecules such as collagens and glycoproteins that provide structural and biochemical support to surrounding cells (Younesi et al, 2024). In addition, fibroblasts produce and degrade essential growth factors and cytokines involved in inflammatory cellular responses. These cells actively migrate and are crucial for maintaining tissue-specific homeostasis by forming dynamic networks of junctions with surrounding cells. In the heart, cardiac fibroblasts are essential for preserving myocardial structural integrity and coordinating reparative or regenerative responses following injury (Forte et al, 2020). Through the production of extracellular matrix components and the secretion of cardiogenic factors such as FGF-1, they influence cardiomyocyte behavior and tissue remodeling (Palmen et al, 2004). Notably, fibroblasts can also form functional gap junctions with cardiomyocytes via connexin-43, enabling electrical coupling that modulates impulse conduction, particularly in the infarct border zone (Ongstad and Kohl, 2016). These structural and electrical interactions support normal cardiac function, and their disruption may contribute to arrhythmogenesis and other pathological phenotypes. Given their widespread presence throughout the human body, understanding the behavior of fibroblasts in tissues has been extensively investigated. Due to their ease of isolation and culture, fibroblasts have been extensively studied in vitro in both basic and clinical research

[1]Department of Pharmaceutical Sciences, Laboratory of Physiopharmacology, University of Antwerp, Universiteitsplein 1, 2610 Antwerp, Belgium. [2]Center of Excellence Infla-Med, University of Antwerp, Antwerp, Belgium. [3]Faculty of Medicine, Department of Pathology and Immunology, University of Geneva, Geneva, Switzerland. ✉E-mail: leonardo.martin@uantwerpen.be
https://doi.org/10.1038/s44321-025-00283-7 | Published online: 31 July 2025

(Gomes et al, 2021). Initial studies focused on the dynamics of fibroblasts in tissue repair, particularly in relation to dermal wound healing. These studies showed that fibroblasts need to be activated to proliferate and migrate under specific pathophysiological conditions, such as wound healing and fibrosis, thereby playing a fundamental role in tissue development and repair (Gabbiani, 2021). Therefore, research into the mechanisms of fibroblast activation, phenotypic transition and migration in the contexts of injury response, tissue regeneration, wound healing and fibrosis contributes to our understanding of the regenerative capacity of the human body.

The mechanisms that govern the activation of fibroblasts involve a complex interplay of physical and biochemical factors triggered by tissue stress, such as acute injuries (Tomasek et al, 2002; Hinz and Gabbiani, 2010). One of the initial responses to tissue stress is the modification of the ECM, leading to tissue stiffening and disruption of mechanical homeostasis. This change is accompanied by the release of inflammatory signals, notably TGF-β and tumor necrosis factor-alpha (TNF-α). These signals induce cytoskeletal remodeling, which in turn alters the forces generated by cells and their mechanical properties. Persistent injury shifts the response from wound healing to fibrosis, with fibroblasts undergoing a phenotypical transition to myofibroblasts, driven by the TGF-β pathway. This transition is marked by the production of alpha-smooth muscle actin (α-SMA) and the increased deposition of ECM components, such as collagen type I, altering the tissue microenvironment (D'Urso and Kurniawan, 2020) (Fig. 1A).

The activation of myofibroblasts is significantly modulated by mechanical stimuli, such as matrix stiffness, a characteristic frequently observed in aging tissues, which facilitates their activation. Early studies on mechanical stimuli focused on dermal wound healing and demonstrated that reducing mechanical stress during the healing process inactivated myofibroblasts, causing them to enter a quiescent state or undergo apoptosis. Conversely, chronic mechanical stress or recurrent injuries sustain myofibroblast activation, impeding the healing process and promoting fibrosis (Bochaton-Piallat et al, 2016). The dynamic interplay between fibroblasts and the extracellular matrix (ECM), where cells both modify and respond to the mechanical and physical properties of their environment, governs various cellular behaviors through mechanotransduction (D'Urso and Kurniawan, 2020). Understanding how these physical and mechanical stimuli impact fibroblast phenotype transitions is crucial for developing strategies to counteract fibrotic responses and effectively treat fibrotic diseases.

## Cardiac fibrosis

Cardiac fibrosis is a pathological manifestation observed in a variety of conditions, encompassing ischemic and nonischemic heart failure (HF), genetic cardiomyopathies, diabetes mellitus, and the aging process. Cardiac fibroblasts, which constitute ~10–20% of the total cell population of the heart, are the primary cells involved in the myocardial fibrotic response, as evidenced by the correlative expression of ECM proteins. Fibrotic responses are initiated by increased levels of circulating and myocardial fibrosis-promoting growth factors and cytokines. These factors bind to surface receptors on fibroblasts, including receptors for cytokines such as TNF-α, as well as CD44, syndecans, and integrins, activating signaling pathways and transcription factors including Smads, MAPKs, NF-κB, and AKT. This activation process prompts the transformation of cardiac fibroblasts into myofibroblasts, which express α-SMA and produce tissue inhibitors of metalloproteinases and metalloproteinases (TIMPs and MMPs, respectively) to modulate ECM homeostasis (Gibb et al, 2020). The synthesis and release of fibrosis-promoting growth factors and cytokines in cardiac fibroblasts are regulated by these transcription factors. In addition, the growth factors and cytokines secreted from cardiac fibroblasts, cardiomyocytes, and endothelial cells create a positive feedback loop, thereby amplifying the fibrotic response (Hall et al, 2021).

## Skeletal muscle fibrosis

Muscle fibrosis frequently occurs in muscular dystrophies, aging, and post-injury scenarios. It compromises both the functional and structural properties of skeletal muscle and impedes muscle fiber regeneration following injury. Furthermore, fibrosis heightens the muscle's susceptibility to re-injury (Huard et al, 2002). Muscle fibrosis is closely associated with inflammation, a relationship that is particularly evident in commonly used experimental models—such as myotoxin injection or freeze injury—that trigger robust inflammatory responses during muscle repair. Following muscle injury, neutrophils are mobilized to the injury site to remove tissue debris and initiate the regenerative process (Silvestre-Roig et al, 2019). Neutrophils have the necessary components for fibrotic scar tissue and ECM transport (Fischer et al, 2022). They release chemoattractant cytokines, further promoting the infiltration of monocytes and macrophages. Macrophages, which exhibit heterogeneous phenotypes, play a significant role in both muscle fibrosis and regeneration (Martins et al, 2020). Classically activated M1 macrophages release pro-inflammatory cytokines such as TNF-α and interleukin-6 (IL-6), which promote fibroblast proliferation. In contrast, alternatively activated M2 macrophages secrete transforming growth factor-β1 (TGF-β1) and fibronectin. An imbalance between the activation of M1 and M2 macrophages results in elevated TGF-β1 expression, which prevents the apoptosis of fibro/adipogenic progenitors (FAPs)—a key population of resident mesenchymal stromal-like cells in skeletal muscle—thereby promoting their differentiation into fibrogenic cells and leading to excessive extracellular matrix (ECM) deposition and fibrosis. In addition, platelet-derived growth factor receptor beta (PDGFRβ)+ profibrotic mesenchymal cells, which overlap with PDGFRα+ cells in fibrotic muscle, proliferate in response to injury and transdifferentiate into myofibroblasts through the activation of integrin αv (Lu et al, 2022; Mahdy, 2019). It has also been shown that mesenchymal cells expressing PDGFRβ mediate fibrosis in both skeletal muscle and the heart through a mechanism involving the αv integrin, and that αv integrin inhibitors attenuate fibrotic responses in animal models (Murray et al, 2017).

## SMAD7's pathways of action

The TGF-β signaling cascade, central to fibrotic responses, begins when TGF-β ligands bind to type II receptors (TβRII), which recruit and phosphorylate type I receptors (TβRI). Upon activation, TβRI phosphorylates receptor-regulated Smads (R-Smads), namely Smad2 and Smad3. These phosphorylated R-Smads then form heteromeric complexes with the common mediator Smad4,

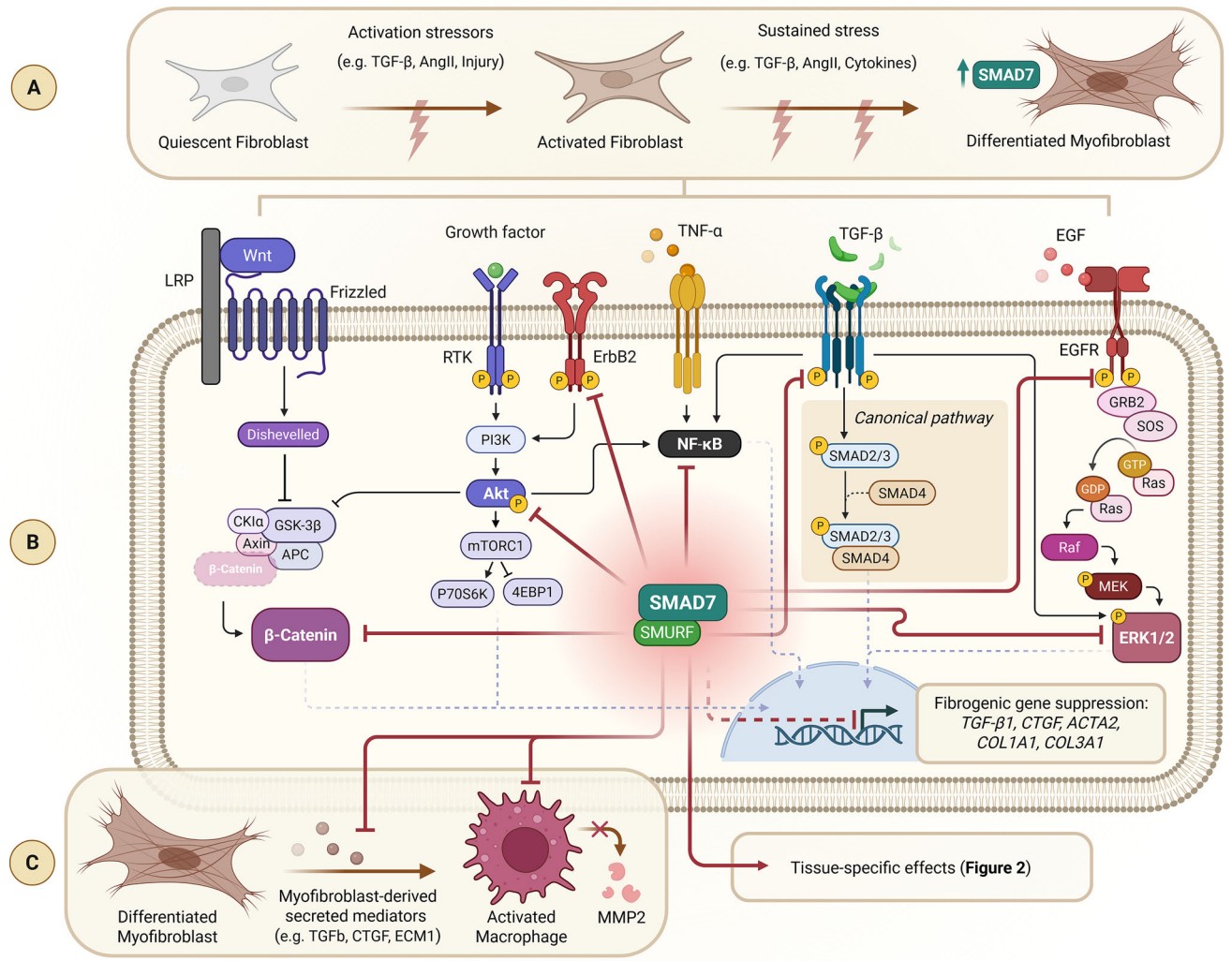

**Figure 1. SMAD7's cascading effects on myofibroblast function regulate the repair and remodeling of cardiac, skeletal, and vessel tissue.**

(A) Fibroblast-to-myofibroblast differentiation involves TGF-β1/Smad signaling activation, cell-ECM mechanotransduction, and synthesis of modulators in the cell surface, like fibronectin. This process includes an intermediate proto-myofibroblast stage characterized by increased proliferation, migration, and stress fiber formation, leading to mature myofibroblasts with enhanced contractile strength and extensive ECM interactions. (B) SMAD7 modulates and is modulated by various signaling pathways. In the Wnt pathway, SMAD7 influences β-catenin activity and stability by acting as a SMURF adaptor. It also affects NF-κB signaling by inhibiting IκB phosphorylation and degradation, thereby disrupting the recruitment of components involved in downstream cell death pathways. In addition, SMAD7 expression is upregulated by inflammatory cytokines such as TNF-α and IFN-γ, which are key effectors in pathways controlling cell fate, including the Hippo signaling pathway. (C) Limitation of macrophage activation by smad7-controlled release of fibrogenic substrates in myofibroblasts and metalloproteinase-2 (MMP2) inhibition. SMAD mothers against decapentaplegic homolog 7, SMURF Smad ubiquitination regulatory factors, TGF-β transforming growth factor beta, Ang II angiotensin II, NF-κB nuclear factor kappa B, TNF tumor necrosis factor, EGF epidermal growth factor, RTK receptor tyrosine kinase, EGFR epidermal growth factor receptor, ErbB2 epidermal growth factor receptor 2, Akt protein kinase B, mTORC1 mechanistic target of rapamycin complex 1, ERK1/2 extracellular signal-regulated kinase 1/2, CTGF connective tissue growth factor, COL1A1 collagen Type I Alpha 1 Chain, COL3A1 collagen Type III Alpha 1 Chain, ACTA2 alpha-smooth muscle actin, ECM extracellular matrix.

enabling their translocation into the nucleus. Once in the nucleus, Smad complexes regulate the transcription of genes encoding fibrotic mediators such as collagen type I alpha 1 (COL1A1), connective tissue growth factor (CTGF), alpha-smooth muscle actin (α-SMA), and other extracellular matrix (ECM) components. The activation of this pathway is

tightly regulated to ensure controlled tissue repair and to prevent pathological fibrosis. Inhibitory Smads (I-Smads), including SMAD7 and, to a lesser extent, SMAD6, play a pivotal role in maintaining this balance (Schmierer and Hill, 2007).

Biochemically, SMAD7 functions as a negative feedback regulator of the TGF-β

pathway by directly binding to activated TβRI, thereby preventing the phosphorylation and activation of R-Smads. In addition, SMAD7 recruits E3 ubiquitin ligases, notably SMURF1 and SMURF2, to the receptor complex, facilitating the ubiquitination and subsequent proteasomal degradation of TβRI. Through these mechanisms, SMAD7

interrupts the fibrotic signaling cascade at multiple levels, effectively limiting ECM accumulation and tissue scarring.

At the molecular level, SMAD7 interacts with several critical nodes within the TGF-β signaling pathway. It inhibits R-Smad activation by competitively binding to TβRI and preventing Smad2/3 phosphorylation. SMAD7 also enhances TβRI turnover via SMURF-mediated ubiquitination and impedes the nuclear translocation and transcriptional activity of phosphorylated Smad2/3-Smad4 complexes. Furthermore, it can interfere with the recruitment of transcriptional co-activators necessary for the expression of profibrotic genes. These multilayered interactions collectively suppress TGF-β-driven profibrotic responses (de Ceuninck van Capelle et al, 2020).

In addition to its canonical inhibitory role, SMAD7 engages in crosstalk with multiple non-canonical pathways—such as NF-κB, Wnt/β-catenin, and MAPK—thereby integrating diverse inflammatory and profibrotic stimuli. Specifically, SMAD7 interferes with NF-κB activation by inhibiting IκB degradation (Bitzer et al, 2000), regulates Wnt signaling through modulation of β-catenin stability via SMURF-mediated ubiquitination (Gao et al, 2009), and modulates MAPK pathways, including p38 and ERK1/2, in TGF-β-driven contexts (Edlund et al, 2002). Notably, while SMAD6 shares structural similarities with SMAD7, its primary role lies in modulating BMP signaling, and it is less implicated in adult fibrotic pathologies (Ishisaki et al, 1999). The broader regulatory functions of SMAD7, including its influence on cytokine responsiveness and immune cell modulation, further underscore its therapeutic relevance in tissue remodeling and fibrosis (de Ceuninck van Capelle et al, 2020).

These multifaceted actions position SMAD7 as a versatile regulator within fibrotic and inflammatory networks (Fig. 1B).

## Role of SMAD7 on biological mechanisms in cardiac, skeletal, and vessel tissues

Several studies have highlighted the critical role of SMAD7 in regulating diverse biological processes across cardiac, skeletal muscle, and vascular tissues (Table 1; Fig. 2). In cardiac tissues, SMAD7 has shown antifibrotic and cardioprotective effects in models of myocardial infarction (Yuan et al, 2017; Su et al, 2024), hypertrophic obstructive cardiomyopathy (Zhang et al, 2023), heart failure (Humeres et al, 2024), and diabetic cardiomyopathy (Meng et al, 2019; Li et al, 2020). In skeletal muscle, SMAD7 has been shown to reduce fibrosis and enhance regeneration in dystrophic and injury-induced models (Kepreotis et al, 2024; Kobayashi et al, 2016), and to prevent muscle wasting in cachexia via ActRIIB signaling inhibition (Winbanks et al, 2016; Maricelli et al, 2018).

At the functional level, SMAD7 modulates fibrosis primarily by suppressing fibrogenic gene expression, inhibiting myofibroblast activation, and controlling inflammatory responses. These effects are observed in models of pressure overload-induced cardiac remodeling (Humeres et al, 2022), high-glucose-induced cardiac fibroblast activation (Che et al, 2020), and myoblast-to-fibroblast transition in skeletal muscle (Song et al, 2024). These actions occur through both TGF-β-Smad-dependent and Smad-independent pathways, including regulation of NF-κB, MAPK, TNF, and Akt/mTOR signaling axes (Che et al, 2020; Li et al, 2022; Kepreotis et al, 2024; Tao et al, 2019; Humeres et al, 2024, 2022). In vascular tissues, although less studied, SMAD7 has been shown to attenuate vascular fibrosis by suppressing vascular smooth muscle cell (VSMC) proliferation and reducing the expression of profibrotic mediators (Rodríguez-Vita et al, 2005; Wang et al, 2021).

Recent studies have elucidated the importance of cell-specific SMAD7 expression patterns. In cardiac tissue, SMAD7 is upregulated in myofibroblasts, cardiomyocytes, and infiltrating immune cells at the infarct border and core, acting as an endogenous feedback mechanism to limit TGF-β-driven fibrosis (Humeres et al, 2022; Li et al, 2022). In skeletal muscle, SMAD7 expression increases in activated myoblasts and fibroblasts within injury sites, contributing to enhanced regeneration and reduced ECM accumulation (Kepreotis et al, 2024; Song et al, 2024).

Genetic manipulation approaches, including transgenic mouse models and viral vector-based overexpression or knock-down systems, have further clarified SMAD7's functional role. For example, myofibroblast-specific deletion of SMAD7 in the heart leads to exaggerated fibrosis, increased MMP2 activity, and enhanced macrophage proliferation and activation (Humeres et al, 2024). Conversely, SMAD7 overexpression limits ECM remodeling, suppresses fibrogenic macrophage polarization, and improves cardiac and skeletal muscle function (Su et al, 2024; Winbanks et al, 2016; Kobayashi et al, 2016). Notably, these antifibrotic effects occur through both inhibition of Smad2/3 signaling and attenuation of non-canonical profibrotic pathways (e.g., ErbB2, MAPK), without disrupting TGF-β receptor activity (Humeres et al, 2022; de Ceuninck van Capelle et al, 2020).

Recent work by Humeres et al (2024) demonstrated that myofibroblast-specific deletion of Smad7 worsens systolic dysfunction and accelerates diastolic dysfunction following pressure overload, driven by increased fibrosis, MMP2 activity, and macrophage proliferation. Secretomic analysis indicated that loss of Smad7 promotes the secretion of structural collagens, matricellular proteins, and TGF-β-inducing factors, further amplifying the fibrotic response (Humeres et al, 2024). Another study revealed that, while macrophage-specific Smad7 induction occurs after myocardial infarction, its functional role in repair is limited compared to the profound antifibrotic impact of myofibroblast-specific Smad7 (Li et al, 2022).

Although less studied than in cardiac and skeletal muscle tissues, SMAD7 also modulates fibrotic processes in the vasculature. Vascular fibrosis is typically initiated by endothelial dysfunction and inflammatory activation, leading to vascular smooth muscle cell (VSMC) phenotypic switching and excessive ECM deposition. In this context, SMAD7 exerts antifibrotic effects by counteracting both canonical TGF-β/Smad2/3 signaling and non-canonical pathways. In VSMCs, SMAD7 overexpression preserves the contractile phenotype and suppresses hyperproliferation, as demonstrated in BMP9/10–SMAD7 axis studies (Wang et al, 2021). Moreover, in endothelial cells, SMAD7 upregulation downstream of VEGF165 attenuates profibrotic DLL4/Notch4 signaling (Lv et al, 2019). Intriguingly, SMAD7 also inhibits angiotensin II-induced fibrosis in a TGF-β-independent manner via p38 MAPK, further emphasizing its broader regulatory reach (Rodríguez-Vita et al, 2005).

Cumulatively, these findings underscore the critical importance of SMAD7 as a central regulator of tissue repair, fibrosis, and immune modulation across multiple

**Table 1. Overview of studies on SMAD7 targeting and corresponding end-points linked to biological mechanisms in cardiac, skeletal, and vessel tissue.**

| Mechanism | Approach | Model | Outcome (following SMAD7 modulation) | Pathway | Reference |
|---|---|---|---|---|---|
| Cardiac fibrosis | In vitro; in vivo | CFs and macrophages (in vitro); Smad7 KO mice; pressure overload-induced cardiac hypertrophy and HF mice model (in vivo). | Loss of SMAD7 promoted cardiac fibrosis, increased MMP2 activity, and enhanced macrophage activation. | TGFβ-Smad2/3-dependent and TGFβ-Smad2/3-independent (ErbB2 and GDF15/GFRAL) | Humeres et al, 2024 |
| | In vivo | Aged mdx/utrn haploinsufficient (+/−) mice. | SMAD7 expression alleviated interstitial fibrosis and restored cardiac function in dystrophic hearts. | TGFβ-Smad2/3-dependent with associated downregulation of MAPK (ERK1/2 and p38) | Kepreotis et al, 2024 |
| | In vitro: in vivo | Neonatal rat cardiac fibroblasts (NRCFs) and neonatal rat cardiac myocytes (NRCMs) (In vitro); MI LAD ligation-induced mice model (in vivo). | CAR3 stabilized SMAD7 and attenuated TGF-β/Smad2/3 signaling, thereby reducing myofibroblast differentiation. | TGFβ-Smad2/3-dependent | Su et al, 2024 |
| | In situ; in vivo | Retrospective study—septal myocardium samples (in situ); Pediatric and adult patients with Hypertrophic obstructive cardiomyopathy (HOCM) (in vivo). | Reduced SMAD7 expression was observed in fibrotic regions of HOCM patient myocardium, suggesting loss of SMAD7 may contribute to disease pathology. | TGFβ-Smad2/3-dependent | Zhang et al, 2023 |
| | In vitro; in vivo | Myofibroblasts (in vitro); myofibroblast-specific Smad7 loss (MFS7KO) mice; Nonreperfused MI mouse model (in vivo). | SMAD7 knockout in myofibroblasts elevated collagen I and fibronectin levels, enhancing fibrotic remodeling. | TGFβ-Smad2/3-dependent and TGFβ-Smad2/3-independent (ErbB1/ErbB2) | Humeres et al, 2022 |
| | In vitro; in vivo | CFs (in vitro); Transverse aortic constriction (TAC) mouse model (in vivo). | NEAT1 epigenetically suppresses SMAD7 via EZH2, promoting fibrosis and cardiac dysfunction. | TGFβ-Smad2/3-dependent | Ge et al, 2022 |
| | In vitro; in vivo | Myeloid cell-specific loss of Smad7 mice (MyS7KO) (in vitro); Smad7 KO mice (in vivo). | SMAD7 deletion in myeloid cells hada limited effect on post-MI fibrosis, suggesting a minor role for macrophage-derived SMAD7 in fibrotic remodeling. | TGFβ-Smad2/3-dependent and TGFβ-Smad2/3-independent (TNF/NF-κB) | Li et al, 2022 |
| | In vitro; in vivo | Neonatal rat cardiac fibroblasts (NRCFs). | miR-96-5p suppresses SMAD7, enhancing Smad3 activation and fibroblast proliferation. | TGFβ-Smad2/3-dependent | Gu et al, 2022 |
| | in vitro | Human cardiac fibroblasts (HCFs). | SMAD7 upregulation inhibited myofibroblast transition and mitigated hypertensive fibrosis. | TGFβ-Smad2/3-dependent | Xiao et al, 2020 |
| | In vivo | T1DM STZ-induced mice. | SMAD7 expression suppressed EndMT and fibrotic remodeling in diabetic myocardium. | TGFβ-Smad2/3-dependent; miR-21/Smad7 axis | Li et al, 2020 |
| | In vitro; in vivo | In vivo: Rat MI model via LAD ligation (observational: ↓SMAD7 in fibrotic myocardium) In vitro: cardiac fibroblasts (CFs) treated with angiotensin II (functional: SMAD7 knockdown using siRNA). | SMAD7 was downregulated in MI myocardium. In vitro knockdown of SMAD7 in CFs enhanced fibrotic and inflammatory gene expression. | TGF-β/Smad2/3-dependent; NF-κB activation upon SMAD7 silencing | Chen et al, 2020 |
| | In vivo | CCl4-induced hepatic fibrosis in rats; hepatic stellate cells (HSCs) in vitro. | Silymarin treatment reduced liver fibrosis by upregulating SMAD7 and suppressing TGF-β/Smad2/3 signaling. | TGF-β/Smad2/3-dependent (via SMAD7 induction) | Meng et al, 2019 |
| | In vivo | Patients with chronic heart failure (CHF) by dilated cardiomyopathy (DCM) and ischemic cardiomyopathy (ICM). | SMAD7 activity suppressed fibrogenic signaling and promoted myocardial remodeling in CHF patients. | TGFβ-Smad2/3-dependent and TGFβ-Smad2/3-independent (Akt/mTOR) | Tao et al, 2019 |
| | In vitro | CFs from neonatal SD rats. | SMAD7 expression reduced inflammatory responses and ECM production in high-glucose conditions. | TGFβ-Smad2/3-dependent and TGFβ-Smad2/3-independent (NF-κB) | Che et al, 2020 |
| | In vitro | CFs from neonatal SD rats. | SMAD7 overexpression attenuated collagen synthesis and fibrotic gene expression in cultured fibroblasts. | TGFβ-Smad-dependent | Zhang et al, 2018 |
| | In vitro; in vivo | CFs (in vitro); MI LAD ligation-induced mice model (In vivo). | SMAD7 reinstatement reduced infarct-induced fibrosis and ECM accumulation. | TGFβ-Smad2/3-dependent | Yuan et al, 2017 |
| | In vitro; in vivo | CFs (in vitro); Atrial fibrillation (AF)-induced rabbits (in vivo). | SMAD7 activity mitigated atrial fibrosis in fibrillation models. | TGFβ-Smad-dependent | He et al, 2016 |
| | In vivo | Smad7 KO; Ang II-induced hypertension. | SMAD7 preserved myocardial structure and limited fibrotic gene expression in hypertensive remodeling. | Sp1-TGFβ/smad3-NF-κB | Wei et al, 2013a, 2013b |
| | In vitro; in vivo | CFs and COS-7 fibroblasts (in vitro); MI rat model (in vivo). | Post-MI cardiac fibrosis correlated with reduced SMAD7 expression | TGFβ-Smad2/3-dependent | Wang et al, 2002 |
| | In vivo | MI LAC ligation-induced rats. | SMAD7 enhancement promoted cardiovascular maturation and reduced cardiomyocyte apoptosis. | TGFβ-Smad2/3-dependent | Hao et al, 2000 |

**Table 1.** (continued)

| Mechanism | Approach | Model | Outcome (following SMAD7 modulation) | Pathway | Reference |
|---|---|---|---|---|---|
| Cardiac development and function | In vivo | Smad7 KO mice. | SMAD7 knockout led to cardiovascular defects and impaired apoptosis regulation during development. | TGFβ-Smad2/3-dependent | Chen et al, 2009 |
| Skeletal muscle growth and differentiation (myogenesis) | In vitro | C2C12 myoblasts. | SMAD7 modulated myogenesis via β-catenin signaling. | TGFβ signaling independent | Tripathi et al, 2019 |
| | In vitro | Bovine skeletal muscle cells. | SMAD7 regulated muscle differentiation through SNIP1 interaction. | TGFβ-Smad-dependent | Yang et al, 2019 |
| | In vitro | Pre and postnatal skeletal muscle cells from Yorkshire pigs. | SMAD7 promoted prenatal and postnatal skeletal muscle growth. | TGFβ-Smad2/3-dependent | Hua et al, 2016 |
| | In vitro; in vivo | Myoblast cultures (in vitro); SMAD7 knockout mice with cardiotoxin injury (in vivo). | SMAD7 deletion impaired myogenic differentiation and muscle regeneration, associated with sustained TGF-β/Smad2/3 activation. | TGF-β/Smad2/3-dependent | Cohen et al, 2015 |
| | In vitro | C2C12 or ovine primary myoblasts. | SMAD7 governed myostatin feedback in myogenic cell types. | TGFβ-Smad2/3-dependent | Forbes et al, 2006 |
| | In vitro | C2C12 myoblasts. | SMAD7 regulated MyoD expression during myogenic differentiation. | TGFβ-Smad2/3-dependent | Kollias et al, 2006 |
| Muscle mass | In vitro; in vivo | Ventricular myocytes; embryonic fibroblasts (in vitro); C57BL/6J mice (In vivo). | SMAD7 upregulation prevented Thbs1-driven muscle wasting. | TGFβ-Smad2/3-ATF4 axis | Vanhoutte et al, 2024 |
| | In vivo | C57BL/6 mice. | SMAD7 expression improved physical performance through enhanced muscle mass. | Activin A (ActRIIB) | Maricelli et al, 2018 |
| | In vivo | CD2F1 or BALB/c mice. | SMAD7 gene therapy prevented cancer-induced cachexia. | Activin A (ActRIIB) | Winbanks et al, 2016 |
| Skeletal muscle fibrosis | In vivo | Aged mdx/utrn haploinsufficient (+/−) mice. | SMAD7 expression reduced ECM accumulation and improved muscle performance in dystrophic models. | TGFβ-Smad2/3-dependent | Kepreotis et al, 2024 |
| | In vitro; in vivo | C2C12 myoblasts (in vitro) and C57BL/6 mice (in vivo). | SMAD7 controlled myoblast-to-fibroblast transition and attenuated muscle fibrosis. | TGFβ-Smad2/3-dependent | Song et al, 2024 |
| | In vivo | Kinin (Bdkrb1/ Bdkrb2) receptor KO mice; muscle contusion mouse model. | SMAD7-mediated response potentially linked to kallikrein-kinin pathway in trauma-induced muscle repair. | TGFβ-Smad2/3-dependent | Martins et al, 2023 |
| | In vivo | C57BL/6 mice MDSCs-transplanted; muscle contusion mouse model. | SMAD7 combined with MyoD signaling enhanced regeneration and reduced fibrosis in injured muscle. | TGFβ-Smad2/3-dependent | Kobayashi et al, 2016 |
| Vascularization | In vitro; in vivo | Several cell type-specific loss- and gain-of-function (in vitro); BMP10-LacZ reporter, BMP10loxP//loxP, BMP9/ 10dko, ROSA26iBMP10, and ROSA26iSmad7 transgenic mice (in vivo). | SMAD7 suppressed VSMC proliferation and preserved vascular contractile phenotype. | BMP10-ALK1-SMAD1/5-dependent (TGF-β-independent) | Wang et al, 2021 |
| | In vitro; in vivo | Endometrial stromal cells from human endometrium (in vitro); Double transgenic (VEGFtetO/tetO/β-actin-tetR-Krab) mice (in vivo). | SMAD7 mediated VEGF165 antifibrotic effects in endometrial stromal cells via Notch4 signaling. | DLL4/Notch4/Smad7 | Lv et al, 2019 |
| Vascular fibrosis | In vivo | Systemic infusion of Ang II in Wistar rats. | SMAD7 activity downregulated CTGF expression and profibrotic signaling in Ang II-induced vascular remodeling. | TGF-β-independent; p38 MAPK-mediated | Rodríguez-Vita et al, 2005 |

*Smad7* mothers against decapentaplegic homolog 7, *MI* myocardial infarction, *Ang II* angiotensi II, *CFs* cardiac fibroblasts, *LAC* left coronary artery, *KO* knockout, *HF* heart failure, *LAD* left anterior descending, *SD* Sprague-Dawley, double mutant (targeted/spontaneous mutation), *mdx/utrn* mouse strain with severe dystrophic phenotype, *MMP2* matrix metalloproteinase-2, *CAR3* carbonic anhydrase 3, *EndMT* endothelial-to-mesenchymal transition, *NEAT1* Nuclear Paraspeckle Assembly Transcript 1, *SNIP1* Smad Nuclear Interacting Protein 1, *Thbs1* Thrombospondin 1, *MDSCs* muscle-derived stem cells, *VEGFI65* vascular endothelial growth factor, *ErbB1* epidermal growth factor receptor, *ErbB2* epidermal growth factor receptor 2, *Sp1* transcription factor Sp1, *NF-κB* nuclear factor kappa B, *TNF* tumor necrosis factor, *GDF15* growth differentiation factor 15, *GFRAL* GDNF receptor-alpha-like, *MAPK* mitogen-activated protein kinase, *Akt* protein kinase B, *mTOR* mammalian target of rapamycin, *ActRIIB* Activin receptor type IIB, *ATF4* activating transcription factor 4, *DLL4* delta-like 4, *VSMC* vascular smooth muscle cell, *Notch4* neurogenic locus notch homolog 4, *BMP* bone morphogenetic protein, *ALK* anaplastic lymphoma kinase.

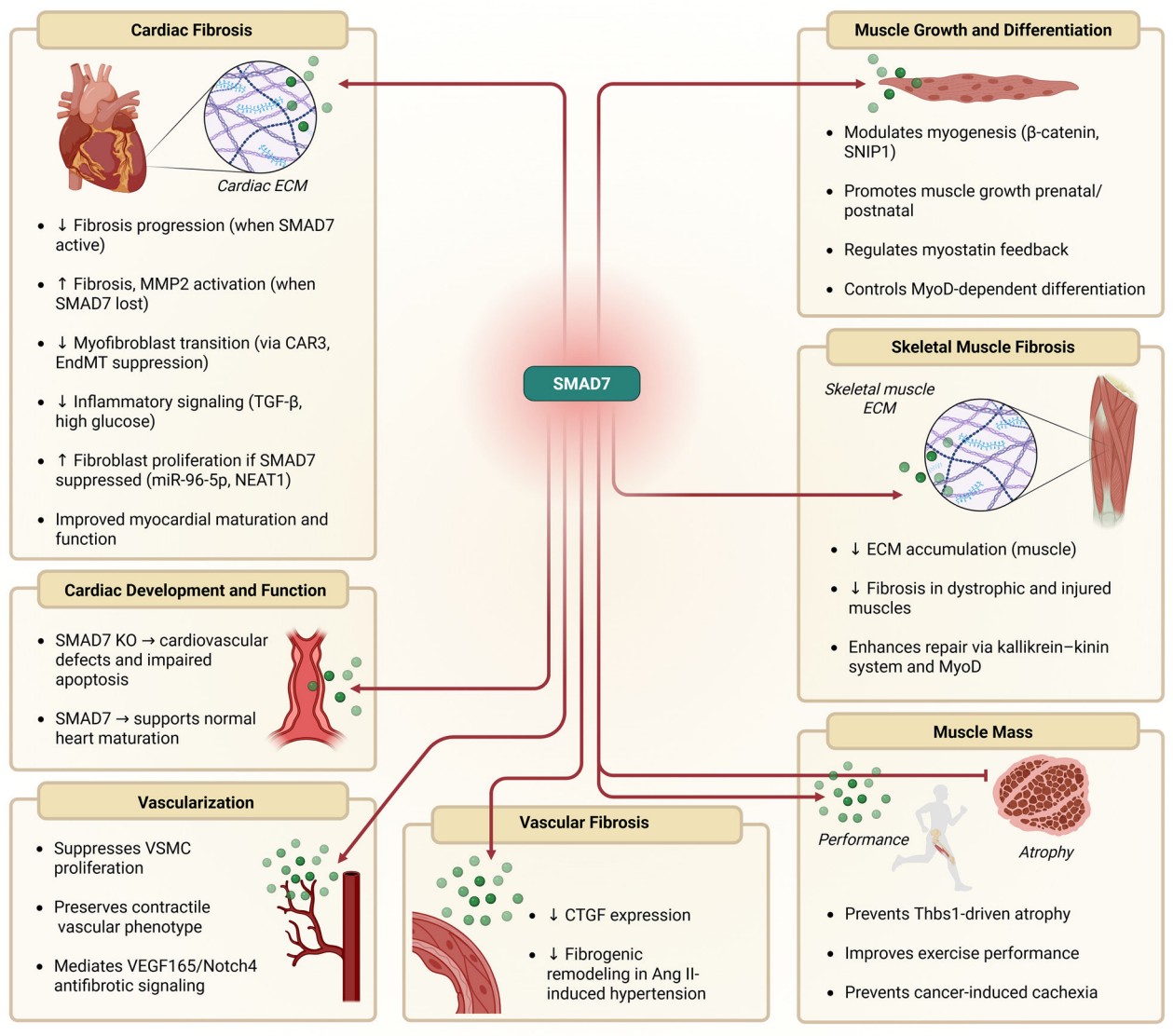

**Figure 2. SMAD7 targeting and corresponding end-points linked to biological mechanisms in cardiac, skeletal, and vessel tissue.**

SMAD mothers against decapentaplegic homolog 7, MMP2 matrix metallopeptidase 2, CAR3 carbonic anhydrase 3, EndMT endothelial-to-mesenchymal transition, TGF-β transforming growth factor beta, NEAT1 nuclear paraspeckle assembly transcript 1, VSMCs vascular smooth muscle cells, VEGF165 vascular endothelial growth factor isoform 165, Notch4 neurogenic locus notch homolog 4, CTGF connective tissue growth factor, Ang II angiotensin II, SNIP1 Smad Nuclear Interacting Protein 1, MyoD myoblast determination protein 1, Thbs1 Thrombospondin 1, ECM extracellular matrix.

organs. By controlling fibroblast, macrophage, and vascular cell phenotypes, SMAD7 preserves tissue integrity and prevents maladaptive remodeling in cardiac, skeletal muscle, and vascular contexts.

## Clinical targeting of SMAD7

Therapeutic strategies directly targeting SMAD7 remain limited but have been primarily investigated in the context of inflammatory disorders, particularly Crohn's disease (CD). In CD, elevated SMAD7 expression disrupts TGF-β1-mediated immunosuppressive signaling, thereby exacerbating mucosal inflammation and contributing to disease pathology (Santacroce et al, 2022). Preclinical studies demonstrated that antisense oligonucleotide-mediated knockdown of SMAD7 could restore TGF-β1 signaling and reduce intestinal inflammation, leading to the development of Mongersen, an oral SMAD7-targeting antisense therapy. Initial phase I and II clinical trials reported encouraging outcomes, including reductions in inflammatory biomarkers and clinical symptomatology (Monteleone et al, 2015). However, the subsequent phase III trial failed to replicate these results, likely due to challenges such as suboptimal drug formulation or chemical inconsistency during manufacturing, and underscored the need for improved delivery strategies and refined patient stratification methods (Monteleone and Stolfi, 2023).

Beyond its immunomodulatory role in inflammatory diseases, SMAD7's pleiotropic regulatory functions—including suppression of fibroblast proliferation, inhibition of myofibroblast differentiation,

and modulation of non-canonical signaling—underscore its therapeutic relevance in fibrotic pathologies. Given that intestinal fibrosis is a frequent complication of CD, further investigation into SMAD7's dual regulatory capacity in inflammation and fibrotic remodeling is warranted.

Beyond the cardiovascular and musculoskeletal systems, SMAD7 has demonstrated potent antifibrotic effects in other organ contexts. In a murine model of idiopathic pulmonary fibrosis (IPF), transient gene transfer of Smad7 significantly attenuated bleomycin-induced lung fibrosis by blocking TGF-β-mediated signaling and myofibroblast activation (Nakao et al, 1999). Similarly, ocular studies have shown that SMAD7 expression reduces stromal fibrosis and promotes regenerative healing. Specifically, adeno-associated virus-mediated delivery of SMAD7 inhibited corneal scarring in vivo by downregulating profibrotic gene expression (Gupta et al, 2017), while direct overexpression of SMAD7 in alkali-injured corneas accelerated epithelial healing and suppressed fibrotic remodeling (Saika et al, 2005). Mechanistic studies also support this broad antifibrotic capacity: SMAD7 broadly inhibits TGF-β superfamily signaling, including BMP and activin pathways, through receptor-proximal interference, as shown in immune cells and epithelial models (Ishisaki et al, 1999). These findings highlight the broader therapeutic relevance of SMAD7 across diverse fibrotic pathologies.

While therapeutic inhibition of upstream mediators such as TGF-β and SMAD3 has been explored extensively in fibrotic diseases, these approaches are limited by significant off-target effects. TGF-β inhibition interferes with critical homeostatic and immune-regulatory functions, often resulting in impaired tissue repair, endothelial dysfunction, and systemic immune dysregulation (Akhurst and Hata, 2012). Similarly, although SMAD3 plays a central role in driving fibroblast activation and ECM deposition, it is also essential for wound healing and regeneration. Thus, global inhibition of SMAD3 may compromise necessary reparative processes (Biernacka et al, 2015).

In contrast, SMAD7 acts as an intracellular, cell-intrinsic, and context-dependent inhibitor of TGF-β signaling, selectively attenuating profibrotic responses in activated cells while preserving physiological signaling in quiescent tissues (Humeres

et al, 2024; Ishisaki et al, 1999). Its expression is dynamically regulated in response to injury, inflammation, and fibrotic stimuli and is not constitutively active across all cell types. Rather, SMAD7 is upregulated specifically in activated myofibroblasts, myoblasts, and certain immune cell populations at sites of tissue damage (Humeres et al, 2024; Li et al, 2022). This context-dependence allows SMAD7 to locally buffer excessive TGF-β activity without disrupting baseline signaling in homeostatic environments.

As previously described above and summarized in Fig. 1B, SMAD7 does not interfere with ligand–receptor binding but rather inhibits downstream signaling by binding to activated TβRI and promoting its degradation through SMURF-mediated ubiquitination. This mechanistic specificity preserves TGF-β responsiveness in non-activated cells, thereby minimizing unintended immunosuppressive or regenerative impairment. In addition, SMAD7 extends its regulatory influence to non-canonical pathways—such as NF-κB, MAPK, and Wnt—offering a more refined and potentially safer antifibrotic strategy. From a therapeutic perspective, these characteristics make SMAD7 amenable to spatial and temporal control through gene delivery strategies, cell-specific promoters, or ex vivo-engineered cell therapies - such as CAR-T cells or nanoparticle-based platforms. Such approaches may maximize antifibrotic efficacy while avoiding the systemic side effects observed with global TGF-β or SMAD3 blockade.

Collectively, these findings highlight the need for renewed efforts to optimize SMAD7-based therapies using advanced delivery technologies and precision targeting approaches in both inflammatory and fibrotic disease settings.

## What is new on the horizon?

Recent advances in immunotherapy have opened new avenues for targeting fibrosis through cell-based strategies. Chimeric antigen receptor T (CAR-T) cell therapy, traditionally used in oncology, has been adapted to target activated myofibroblasts, key drivers of fibrotic progression (Aghajanian et al, 2019). Using lipid nanoparticle (LNP) technology to deliver modified mRNA encoding fibroblast activation protein (FAP)-specific CARs, Rurik et al developed transient antifibrotic CAR-T cells

capable of selectively clearing myofibroblasts. In mouse models of cardiac injury, these transient CAR-T cells reduced fibrosis, enhanced cardiac function, and minimized the risk of excessive fibroblast depletion, thereby preserving normal wound healing processes (Rurik et al, 2022).

While SMAD7 was not targeted in the initial antifibrotic CAR-T study, recent research in oncology has demonstrated that SMAD7 expression in CAR-T cells enhances their persistence, memory phenotype, and antitumor efficacy by counteracting TGF-β-induced exhaustion (Li et al, 2023; Liang et al, 2024). Mechanistically, SMAD7 modulates TGF-β signaling by inhibiting Smad2/3 phosphorylation and promoting TβRI degradation, reducing CAR-T cell exhaustion while maintaining effector function. Moreover, SMAD7-modified CAR-T cells display enhanced survival and cytokine regulation, balancing therapeutic potency with reduced systemic toxicity.

Given that TGF-β is also a central mediator of fibroblast activation and fibrosis, these findings suggest broader potential applications for SMAD7-modified cellular therapies beyond oncology. By modulating fibroblast activation and immune responses simultaneously, SMAD7 offers a promising strategy for restoring tissue homeostasis in fibrotic diseases such as cardiac fibrosis. Future investigations exploring SMAD7-based genetic modifications in antifibrotic CAR-T or other cell-based therapies could provide transformative approaches for treating fibrosis and chronic inflammation.

## Conclusions

Fibrosis remains a critical pathological feature underlying the progression of cardiac, skeletal muscle, and vascular diseases. SMAD7 has emerged as a key intracellular regulator of tissue remodeling, exerting antifibrotic and immunomodulatory effects through inhibition of both canonical TGF-β/Smad signaling and non-canonical pathways. Its cell-specific actions, particularly within myofibroblasts and immune populations, highlight its potential to restrain maladaptive remodeling while preserving regenerative responses.

Although preclinical studies underscore the therapeutic promise of SMAD7, clinical translation remains limited. The failure of Mongersen in Crohn's disease underscores challenges related to drug formulation,

targeted delivery, and patient stratification. However, SMAD7 remains a compelling target due to its intracellular localization, pathway specificity, and broader regulatory scope. Emerging strategies—such as tissue-specific gene therapy, mRNA-based delivery, and incorporation into engineered cell platforms—offer new opportunities to enhance efficacy and safety. Additionally, combination therapies targeting extracellular matrix remodeling or immune modulation may further augment its therapeutic utility.

Advancing SMAD7-based interventions will require a deeper understanding of its tissue- and context-specific functions, as well as improved translational platforms. As such, SMAD7 represents a promising, yet underutilized, candidate in antifibrotic therapy with the potential to restore tissue homeostasis and mitigate disease progression across multiple organ systems.

## Peer review information

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

## Acknowledgements

This project received funding from The Research Foundation—Flanders (FWO) under the umbrella of the Partnership Fostering a European Research Area for Health (ERA4Health-CARDINNOV) (GA N° 101095426 of the EU Horizon Europe Research and Innovation Programme) to LM and GRYDM; Special University Research Fund, BOF, Seal of Excellence grants, University of Antwerpen, Flemish Government (BE) to LM. We apologize to the many scientists responsible for contributions to the investigation of SMAD7 interactions on tissue regeneration that could not be acknowledged owing to limited space. The figures were created with BioRender.

## Author contributions

**Leonardo Martins**: Conceptualization; Supervision; Funding acquisition; Investigation; Visualization; Methodology; Writing—original draft; Writing—review and editing. **Giulio Gabbiani**: Conceptualization; Investigation; Writing—review and editing. **Guido R Y De Meyer**: Conceptualization; Funding acquisition; Investigation; Writing—review and editing.

## Disclosure and competing interests statement

The authors declare the following financial interests/personal relationships which may be considered as potential competing interests: Mast cells as effectors in hemorrhage and acute cardiovascular diseases (MASTer Consortium) approved by the European Granting Authority as part of the ERA4Health Joint Transnational Call "Research targeting development of innovative therapeutic strategies in cardiovascular disease" (CARDINNOV/2023) program. ERA4Health_EU Network (ERA-NET), Funded by the European Union under the Horizon Europe Framework Programme (Horizon_EU), Co-funded by The Research Foundation – Flanders (FWO).

