## [Peer Review File · EMBO Molecular Medicine]

SMAD7: Riding on fibrosis-limiting routes and beyond

Leonardo Martins, Giulio Gabbiani, and Guido De Meyer

Corresponding author(s): Leonardo Martins (leonardo.martin@uantwerpen.be)

Review Timeline:

Submission Date:	21st May 25
Editorial Decision:	16th Jun 25
Revision Received:	22nd Jun 25
Editorial Decision:	14th Jul 25
Revision Received:	15th Jul 25
Accepted:	17th Jul 25

Editor: Lise Roth

Transaction Report:

16th Jun 2025

Dear Dr. Martins,

Thank you for the submission of your article to EMBO Molecular Medicine. We have now received feedback from the experts who agreed to evaluate your manuscript.

As you will see from the reports below, they overall found the article interesting and well written. They nevertheless make several suggestions to improve the interest and impact of your work.

We would therefore welcome a revised version of your manuscript that would address these points. Please attach a covering letter giving details of the way in which you have handled each of the points raised by the referees. A revised manuscript will once again be subject to review, and we cannot guarantee at this stage that the eventual outcome will be favourable.

We do not publish mini-reviews, but we think your manuscript would be suitable as a Perspective:

<https://www.embopress.org/page/journal/17574684/authorguide#perspectiveguide>. Please note that our Perspective format is flexible. As an indication, we recommend no more than 4 figures, 5000 words and 50 references.

Additionally, please address the following editorial requests:

- We can accommodate a maximum of 5 keywords, please adjust accordingly.
- The corresponding author should be defined on the title page.
- Please note that all corresponding authors must have an ORCID identifier.
- Funding should be merged with Acknowledgements.
- Please rename "Competing Interests" to "Disclosure and Competing Interests Statement".
- Please remove the Authors Contributions from the manuscript and use the free text boxes beneath each contributing author's name in our system to add specific details on the author's contribution.
- Please reformat the references to have 10 authors listed before et al. DOIs should be removed.
- The short summary and the list of abbreviations should be removed.
- For the figures, please note:
 1. If there are certain aspects of your figure draft that are based upon assumptions or where the scientific data remains ambiguous, please add a comment so that we can work with you on an accurate depiction. Please ensure the directionality and nature of interactions is presented accurately.
 2. If the figure or single panels of the figure have been adapted from a published figure, please add this information to the figure legend.
 3. Please only re-use figures or parts of a figure if this is essential for understanding the concept communicated. All re-used material must be explicitly cited.
 4. If you use an image data base for scientific iconography (e.g., BioRender), please let us know if you have a license that allows for publication in an academic journal.

Looking forward to receiving your revised manuscript,

With kind regards,

Lise Roth

***** Reviewer's comments *****

Referee #1 (Remarks for Author):

The review by Leonardo Martin et al. identifies SMAD7 as a promising target in combating tissue fibrosis, a subject largely underexplored in current literature. To fully realize its potential, the paper requires significant revision:

1 The authors summarized available data on the role of SMAD7 in muscle, cardiac, and vascular fibrosis. It would be beneficial if the authors also discussed papers where the therapeutic benefits of SMAD7 have been shown in other fibrotic conditions, such as Idiopathic Pulmonary Fibrosis (IPF) or corneal fibrosis, as examples.

2 On page 2, lines 68-71, the authors discussed the role of fibroblasts in maintaining a normal heart rhythm and referenced Gomes et al., 2021. However, the cited paper is a review and does not discuss this specific role of fibroblasts.

3 Generally, throughout the manuscript, the authors frequently referenced other review papers. Ideally, references to original research papers are needed in such instances.

4 On page 3, when discussing cardiac and skeletal muscle fibrosis, the authors stated that skeletal muscle fibrosis is intricately associated with inflammation (lines 137-8). This statement gives the impression that inflammation is higher in muscle fibrosis than in cardiac fibrosis. This might be primarily due to the different experimental models widely used to induce injury in skeletal muscle (e.g., injection of myotoxins or freeze injury) compared to those used for the heart. This point needs clarification or revision.

5 On pages 3-4, lines 148-151, the authors stated that "An imbalance between the activation of M1 and M2 macrophages results in high expression of TGF- β 1, which activates resident fibroblasts and prevents the apoptosis of fibro/adipogenic progenitors (FAPs), leading to their differentiation into a fibrogenic lineage and resulting in excessive deposition of extracellular matrix (ECM) and fibrosis." As FAPs are indeed resident fibroblasts of skeletal muscle, this sentence needs revision to avoid confusion and improve accuracy.

6 On page 4, lines 189-190, the authors discussed the role of SMAD7 in non-canonical pathways such as NF- κ B, Wnt, and MAPK. It would be highly beneficial to reference the original papers demonstrating SMAD7's role in these specific pathways.

7 On page 5, the first few paragraphs discuss the role of SMAD7 in different biological processes, but no papers are referenced here. While the authors may have summarized many of these references in Table 1, it would be better to include them in the text as well for immediate context.

8 On page 5 and in Table 1, the authors referenced the study by Li et al., 2022, stating that SMAD7 depletion in fibroblasts increased post-MI fibrosis but a limited effect was observed in macrophage-specific knockout (as quoted from the table). However, this study only investigated the role of SMAD7 in macrophages, not in fibroblasts. This needs to be corrected.

9 On page 6, lines 269-282, the authors asserted that while therapeutic inhibition of TGF- β and SMAD3 has significant off-target effects, SMAD7-based therapies would be safe. However, considering that SMAD7 plays a significant role in immune modulation (as the authors also discussed), it's important to clarify why modulation of SMAD7 would not incur similar side effects. The authors mentioned that SMAD7 acts as a cell-intrinsic, context-dependent inhibitor of TGF- β signaling, but it needs to be further discussed how this can be therapeutically harnessed. One would expect that global activation of SMAD7 would have the same effect as global inhibition of the TGF- β /SMAD3 pathway. This critical aspect requires more in-depth discussion within the manuscript.

10 Table 1 is a very useful resource, but it contains several mistakes and requires significant revision. A few examples are included below:

- Su et al.'s study (2024) showed that CAR3 stabilizes SMAD7, not the other way around.
- Zhang et al., 2023, did not show that SMAD7 upregulation is associated with reduced myofibroblast fibrosis; rather, they demonstrated that SMAD7 levels are reduced in fibrotic areas.
- The study by Li et al., 2022, as mentioned above, did not investigate the role of SMAD7 in myofibroblasts; they only studied its role in macrophages.
- Rodriguez-Vita et al., 2005, showed that the effect of SMAD7 in vascular fibrosis is, in fact, independent of the TGF- β -SMAD2/3 pathway and is mediated through the P38 MAPK pathway.

11 In Table 1, please provide more details and be accurate on how each study modulates SMAD7. For example, in the study by Meng et al. (2019), they showed that Silymarin reduces fibrotic burden by SMAD7 induction. Also, ensure that the in vitro and in vivo approaches are correctly referenced: for instance, in the study by Chen et al., 2020, SMAD7 targeting was performed only in vitro on CFSSs, and they only reported downregulation of SMAD7 in the MI LAC ligation-induced rat model.

12 Please ensure all relevant studies are included in Table 1. For example, Cohen TV et al. in 2015 showed that genetic disruption of SMAD7 impairs muscle growth, which is not included.

13 Please ensure there are no errors in the figures. For example, in Figure 1b, the effect of Akt on mTORC1 is activation, not

inhibition.

Referee #2 (Remarks for Author):

Fibrosis affects multiple organs including the heart, liver, muscle, lung, and kidney. In this review, authors discussed the expanding role of SMAD7 in cardiac, skeletal muscle, and vascular tissues, emphasizing its promise as a therapeutic target for reprogramming fibrosis. This topic is interesting. However, the following concerns should be addressed.

1. Fibrosis affects multiple organs, liver fibrosis and kidney fibrosis are very common diseases. Why does the author only introduce the expanding role of SMAD7 in cardiac, skeletal muscle, and vascular tissues ?
2. The background introduction is too simple, and directly introducing SMAD7 is too abrupt. It is suggested to introduce the basic knowledge of the SMAD family.
3. The author should also provide a specific introduction to the mechanism of vascular tissues fibrosis.
4. In the "Role of SMAD7 on biological mechanisms in cardiac, skeletal and vessel tissues" part, there are multiple places that do not provide references to support the argument.
5. It is better to reorganise the "Clinical targeting of SMAD7" part, because the description does not match the title very well.

Point-by-point response for manuscript "SMAD7: Riding on fibrosis-limiting routes and beyond" - EMM-2025-21988.

Dear Dr. Lise Roth,

We sincerely thank you for considering our manuscript for publication in EMBO Molecular Medicine. We are also grateful to Referees #1 and #2 for their thoughtful and constructive comments, which significantly helped us improve the quality and clarity of our work.

We have carefully addressed all the points raised by both reviewers and incorporated the recommended changes into a newly revised version of the manuscript. For your convenience, we are submitting the updated manuscript with all modifications clearly highlighted.

To facilitate your evaluation, we have structured our response document as follows:

- (i) Reviewer comments are included in *italicized black text*, and
- (ii) Our detailed responses follow each comment in **bold black text**.

We trust that the revisions have strengthened the manuscript and we look forward to your feedback.

Referee #1:

The review by Leonardo Martin et al. identifies SMAD7 as a promising target in combating tissue fibrosis, a subject largely underexplored in current literature. To fully realize its potential, the paper requires significant revision:

1 The authors summarized available data on the role of SMAD7 in muscle, cardiac, and vascular fibrosis. It would be beneficial if the authors also discussed papers where the therapeutic benefits of SMAD7 have been shown in other fibrotic conditions, such as Idiopathic Pulmonary Fibrosis (IPF) or corneal fibrosis, as examples.

We thank the reviewer for this insightful suggestion. In response, we have now incorporated a new paragraph in the section 'Clinical targeting of SMAD7' highlighting the therapeutic role of SMAD7 in other fibrotic conditions such as idiopathic pulmonary fibrosis and corneal fibrosis. We cite recent in vivo studies demonstrating that SMAD7 overexpression or targeted delivery attenuates fibrosis and enhances tissue repair in both pulmonary and ocular models (Nakao et al., 1999; Saika et al., 2005; Gupta et al., 2017). These additions expand the translational scope of our review and support the broader antifibrotic relevance of SMAD7.

2 On page 2, lines 68-71, the authors discussed the role of fibroblasts in

maintaining a normal heart rhythm and referenced Gomes et al., 2021. However, the cited paper is a review and does not discuss this specific role of fibroblasts.

We thank the reviewer for this important observation. We agree that Gomes et al., 2021 is a general review and does not directly support the specific statement regarding the role of fibroblasts in cardiac electrical conduction. To address this, we have revised the sentence for accuracy and replaced the citation with original studies that directly demonstrate how cardiac fibroblasts influence electrophysiological properties through gap junction formation and electrotonic coupling with cardiomyocytes (e.g., Ongstad & Kohl, 2016). These studies provide experimental evidence supporting the regulatory role of fibroblasts in cardiac rhythm.

3 Generally, throughout the manuscript, the authors frequently referenced other review papers. Ideally, references to original research papers are needed in such instances.

We appreciate the reviewer's important observation. In the revised manuscript, we have systematically reviewed all instances where review papers were cited and, whenever possible, replaced them with the original research articles that generated the referenced findings. This ensures proper attribution and strengthens the evidence base supporting our discussion. While we are aware that the journal recommends a limit of 50 references for perspective manuscripts, we respectfully note that—after incorporating all reviewer suggestions and prioritizing citation of original research—we now retain 72 references. We hope the editorial team will consider this justified, as it serves to enhance the scientific accuracy and depth of the manuscript.

4 On page 3, when discussing cardiac and skeletal muscle fibrosis, the authors stated that skeletal muscle fibrosis is intricately associated with inflammation (lines 137-8). This statement gives the impression that inflammation is higher in muscle fibrosis than in cardiac fibrosis. This might be primarily due to the different experimental models widely used to induce injury in skeletal muscle (e.g., injection of myotoxins or freeze injury) compared to those used for the heart. This point needs clarification or revision.

We appreciate the reviewer's observation and agree that our original phrasing may have implied a disproportionate association of inflammation with skeletal muscle fibrosis compared to cardiac fibrosis. We have revised the sentence to clarify that the prominence of inflammatory responses in skeletal muscle fibrosis is, in part, reflective of the commonly used injury models—such as myotoxin injection or freeze injury—which elicit robust acute inflammation. This clarification better contextualizes the inflammatory component within experimental modeling frameworks and avoids unintended comparison with cardiac fibrosis.

5 On pages 3-4, lines 148-151, the authors stated that "An imbalance between the

activation of M1 and M2 macrophages results in high expression of TGF- β 1, which activates resident fibroblasts and prevents the apoptosis of fibro/adipogenic progenitors (FAPs), leading to their differentiation into a fibrogenic lineage and resulting in excessive deposition of extracellular matrix (ECM) and fibrosis." As FAPs are indeed resident fibroblasts of skeletal muscle, this sentence needs revision to avoid confusion and improve accuracy.

We thank the reviewer for this important clarification. We agree that the original sentence could lead to confusion by suggesting a distinction between resident fibroblasts and fibro/adipogenic progenitors (FAPs), despite FAPs being a major resident fibroblast population in skeletal muscle. We have revised the sentence to more accurately reflect the role of FAPs as resident mesenchymal stromal cells that differentiate into fibrogenic cells in response to persistent TGF- β 1 signaling, thereby contributing to fibrosis.

6 On page 4, lines 189-190, the authors discussed the role of SMAD7 in non-canonical pathways such as NF- κ B, Wnt, and MAPK. It would be highly beneficial to reference the original papers demonstrating SMAD7's role in these specific pathways.

We thank the reviewer for this helpful suggestion. In response, we have added references to the original studies that describe SMAD7's involvement in modulating NF- κ B, Wnt, and MAPK signaling pathways. These citations clarify the mechanistic basis for SMAD7's role beyond canonical TGF- β signaling.

7 On page 5, the first few paragraphs discuss the role of SMAD7 in different biological processes, but no papers are referenced here. While the authors may have summarized many of these references in Table 1, it would be better to include them in the text as well for immediate context.

We thank the reviewer for this important suggestion. To improve readability and provide immediate context, we have added key in-text references corresponding to the studies summarized in Table 1. These references highlight the biological processes regulated by SMAD7 in cardiac, skeletal muscle, and vascular tissues, enhancing the traceability of the described findings.

8 On page 5 and in Table 1, the authors referenced the study by Li et al., 2022, stating that SMAD7 depletion in fibroblasts increased post-MI fibrosis but a limited effect was observed in macrophage-specific knockout (as quoted from the table). However, this study only investigated the role of SMAD7 in macrophages, not in fibroblasts. This needs to be corrected.

We thank the reviewer for pointing out this misattribution. We have corrected the reference to Li et al., 2022, which investigated macrophage-specific SMAD7 deletion and not fibroblast-specific knockout. The description in

Table 1 and the main text has been revised accordingly. We now accurately attribute fibroblast-specific findings to Humeres et al., while Li et al. is cited solely in the context of myeloid-specific knockout experiments.

9 On page 6, lines 269-282, the authors asserted that while therapeutic inhibition of TGF- β and SMAD3 has significant off-target effects, SMAD7-based therapies would be safe. However, considering that SMAD7 plays a significant role in immune modulation (as the authors also discussed), it's important to clarify why modulation of SMAD7 would not incur similar side effects. The authors mentioned that SMAD7 acts as a cell-intrinsic, context-dependent inhibitor of TGF- β signaling, but it needs to be further discussed how this can be therapeutically harnessed. One would expect that global activation of SMAD7 would have the same effect as global inhibition of the TGF- β /SMAD3 pathway. This critical aspect requires more in-depth discussion within the manuscript.

We appreciate the reviewer's critical observation. In response, we have expanded the discussion in the relevant section to clarify how SMAD7's effects differ mechanistically and therapeutically from global inhibition of TGF- β or SMAD3. Specifically, we elaborate on SMAD7's context-specific expression, its intracellular and inducible nature, and how these characteristics can be leveraged to design safer antifibrotic strategies. We now explicitly address how targeted SMAD7 modulation avoids the systemic immunosuppression or repair deficits often seen with TGF- β pathway inhibitors.

10 Table 1 is a very useful resource, but it contains several mistakes and requires significant revision.

A few examples are included below:

- Su et al.'s study (2024) showed that CAR3 stabilizes SMAD7, not the other way around.*
- Zhang et al., 2023, did not show that SMAD7 upregulation is associated with reduced myofibroblast fibrosis; rather, they demonstrated that SMAD7 levels are reduced in fibrotic areas.*
- The study by Li et al., 2022, as mentioned above, did not investigate the role of SMAD7 in myofibroblasts; they only studied its role in macrophages.*
- Rodriguez-Vita et al., 2005, showed that the effect of SMAD7 in vascular fibrosis is, in fact, independent of the TGF- β -SMAD2/3 pathway and is mediated through the P38 MAPK pathway.*

We thank the reviewer for raising critical issues regarding the accuracy of Table 1. In response, we have carefully reviewed all entries and implemented corrections to misassigned pathways, misattributed cellular targets, and incorrect mechanistic conclusions. Specific fixes include clarifying that CAR3 stabilizes SMAD7 (Su et al., 2024), that SMAD7 expression is reduced in fibrotic HOCM myocardium (Zhang et al., 2023), and that Li et al. (2022) investigated macrophage-specific—not fibroblast-specific—Smad7 deletion.

Furthermore, we corrected the pathway attribution for Rodríguez-Vita et al. (2005) to reflect its p38 MAPK dependence and revised several other entries for clarity and accuracy. These corrections ensure the integrity and reliability of Table 1 as a reference for readers.

11 In Table 1, please provide more details and be accurate on how each study modulates SMAD7. For example, in the study by Meng et al. (2019), they showed that Silymarin reduces fibrotic burden by SMAD7 induction. Also, ensure that the in vitro and in vivo approaches are correctly referenced: for instance, in the study by Chen et al., 2020, SMAD7 targeting was performed only in vitro on CFSs, and they only reported downregulation of SMAD7 in the MI LAC ligation-induced rat model.

We thank the reviewer for highlighting the need for more precise descriptions of SMAD7 modulation across studies. In response, we have revised Table 1 to specify how SMAD7 was modulated in each study—whether pharmacologically (e.g., Silymarin-induced in Meng et al., 2019) or genetically (e.g., in vitro siRNA knockdown in Chen et al., 2020). We also clarified which experiments were conducted in vivo versus in vitro to avoid any misinterpretation regarding the experimental scope of each study.

12 Please ensure all relevant studies are included in Table 1. For example, Cohen TV et al. in 2015 showed that genetic disruption of SMAD7 impairs muscle growth, which is not included.

We thank the reviewer for pointing out this omission. We have now added the study by Cohen TV et al. (2015), which demonstrates that genetic disruption of SMAD7 impairs myogenic differentiation and muscle regeneration via sustained TGF- β /Smad2/3 signaling. This study has been incorporated into Table 1, along with model details and mechanistic context.

13 Please ensure there are no errors in the figures. For example, in Figure 1b, the effect of Akt on mTORC1 is activation, not inhibition.

We thank the reviewer for identifying this important oversight. In Figure 1b, we have corrected the interaction between Akt and mTORC1 to accurately reflect Akt's activating effect on mTORC1, in line with established signaling mechanisms. All pathway interactions in the figure were re-checked to ensure consistency with current literature.

Referee #2):

Fibrosis affects multiple organs including the heart, liver, muscle, lung, and kidney. In this review, authors discussed the expanding role of SMAD7 in cardiac, skeletal muscle, and vascular tissues, emphasizing its promise as a therapeutic target for reprogramming fibrosis. This topic is interesting. However, the following concerns should be addressed.

1. Fibrosis affects multiple organs, liver fibrosis and kidney fibrosis are very common diseases. Why does the author only introduce the expanding role of SMAD7 in cardiac, skeletal muscle, and vascular tissues ?

We thank the reviewer for this thoughtful and important comment. We fully agree that SMAD7 plays a well-established antifibrotic role in multiple organ systems, including the liver and kidney, where it modulates TGF- β signaling and limits inflammation-induced matrix accumulation. However, the aim of this Perspective article is to highlight the expanding and comparatively underexplored roles of SMAD7 in striated muscle (cardiac and skeletal) and vascular fibrosis—tissues that have historically received less attention in this context. These systems exhibit unique biomechanical and cellular environments that provide a valuable lens through which to investigate SMAD7's regulatory functions.

In response to the reviewer's suggestion, we have revised the Background section to briefly acknowledge the established antifibrotic roles of SMAD7 in hepatic and renal fibrosis. This addition helps frame the broader biological relevance of SMAD7 while preserving the focus on emerging insights from cardiovascular and musculoskeletal studies.

2. The background introduction is too simple, and directly introducing SMAD7 is too abrupt. It is suggested to introduce the basic knowledge of the SMAD family.

We thank the reviewer for the thoughtful comment. We agree that a basic introduction to the SMAD family is important to better contextualize SMAD7. As suggested, we have now added a brief summary of SMAD family components to the Background section, while maintaining a more detailed explanation in the dedicated "SMAD7's pathways of action" section. This approach ensures conceptual continuity without exceeding the character limit imposed by EMBO Molecular Medicine for Perspective articles.

3. The author should also provide a specific introduction to the mechanism of vascular tissues fibrosis.

We thank the reviewer for highlighting the importance of including a more specific discussion on vascular fibrosis. We fully agree that fibrosis of the vascular system is an important pathological process that merits attention. In the revised manuscript, we have added a concise, focused paragraph under the section "Role of SMAD7 on biological mechanisms in cardiac, skeletal and vessel tissues" to specifically address the mechanisms of vascular fibrosis and the emerging evidence for SMAD7's modulatory role in this context.

Given the format constraints of the Perspective article (limited to 5000 words), we aimed to maintain brevity while ensuring that key mechanistic

insights were captured. The new paragraph integrates findings from three original studies (Wang et al., 2021; Lv et al., 2019; Rodríguez-Vita et al., 2005), which collectively demonstrate SMAD7's context-dependent antifibrotic effects in vascular smooth muscle cells and endothelial cells via both canonical (TGF- β /Smad2/3) and non-canonical (BMP, Notch4, MAPK) pathways. We trust this targeted addition appropriately addresses the reviewer's concern while preserving the manuscript's overall focus and flow.

4. In the "Role of SMAD7 on biological mechanisms in cardiac, skeletal and vessel tissues" part, there are multiple places that do not provide references to support the argument.

We thank the reviewer for this observation. As already addressed in response to Reviewer #1, we have revised this section to ensure that all mechanistic claims are now appropriately supported by original research references. This strengthens the scientific foundation of the discussion and improves citation accuracy throughout the manuscript.

5. It is better to reorganise the "Clinical targeting of SMAD7" part, because the description does not match the title very well.

We thank the reviewer for this constructive suggestion. In response, we have thoroughly revised the "Clinical targeting of SMAD7" section to ensure a more accurate and cohesive alignment between its title and content. The updated section now explicitly discusses therapeutic strategies that have directly targeted SMAD7—such as antisense oligonucleotides in Crohn's disease—while also expanding on the translational relevance of SMAD7's antifibrotic and immunomodulatory roles across multiple organ systems. Moreover, we clarified the mechanistic rationale for targeting SMAD7 in fibrotic and inflammatory disorders and emphasized how its context-specific actions could be leveraged through precision gene delivery technologies. We believe this reorganization improves the coherence and relevance of the section and more clearly frames the potential of SMAD7 as a clinical target.

14th Jul 2025

Dear Dr. Martins,

Thank you for the submission of your revised article to EMBO Molecular Medicine. Both initial referees are satisfied with the revisions, and I will therefore be able to accept your manuscript once the following minor editorial comments are addressed:

1/ please remove the following sections:

- Clinical trial number
- Ethics approval and consent to participate
- Consent for publication
- Availability of Data and Materials

2/ please remove the yellow highlights in the text.

3/ please increase the font in the figures to improve readability. To match the style of the journal, we would also suggest making the overall hue and background of the images whiter (or use lighter tones). In the acknowledgements, please correct to "The figures were created with BioRender".

Looking forward to receiving your revised manuscript,

With kind regards,

Lise Roth

***** Reviewer's comments *****

Referee #1 (Remarks for Author):

suitable for publication

Referee #2 (Remarks for Author):

suitable for publication

The authors addressed the remaining editorial issues.

17th Jul 2025

Dear Dr. Martins,

I am pleased to inform you that your manuscript is accepted for publication and is now being sent to our publisher to be included in the next available issue of EMBO Molecular Medicine.

Your manuscript will be processed for publication by EMBO Press. It will be copy edited and you will receive page proofs prior to publication. Please note that you will be contacted by Springer Nature Author Services to complete licensing information.

This Perspective is free of charge, and we will shortly send you an email with a token. When you are contacted in a few weeks to sign your license agreement and review article proofs, please enter this token into the relevant field in the Springer Nature author services system.

Yours sincerely,

Lise Roth
